# *Fusarium* Species Associated with Maize Leaf Blight in Heilongjiang Province, China

**DOI:** 10.3390/jof8111170

**Published:** 2022-11-06

**Authors:** Xi Xu, Li Zhang, Xilang Yang, Guijin Shen, Shuo Wang, Haolin Teng, Chunbo Yang, Xueyan Liu, Xiangjing Wang, Junwei Zhao, Wensheng Xiang

**Affiliations:** 1Key Laboratory of Agricultural Microbiology of Heilongjiang Province, Northeast Agricultural University, No. 600 Changjiang Road, Xiangfang District, Harbin 150030, China; 2State Key Laboratory for Biology of Plant Diseases and Insect Pests, Institute of Plant Protection, Chinese Academy of Agricultural Sciences, Beijing 100097, China

**Keywords:** *Fusarium* spp., maize, haplotype analysis, genetic diversity

## Abstract

*Fusarium* spp. are among the most important plant pathogens in the world. A survey on maize leaf blight was carried out in Heilongjiang province from 2019 to 2021. Based on morphological characteristics and a phylogenetic analysis on translation elongation factor (*tef1*) and second-largest subunit of RNA polymerase II (*rpb2*) genes, 146 *Fusarium* isolates were obtained and grouped into 14 *Fusarium* species, including *F. ipomoeae* (20.5%), *F. compactum* (17.1%), *F. sporotrichioides* (9.59%), *F. graminearum* (9.59%), *F. citri* (8.9%), *F. asiaticum* (6.85%), *F. verticillioides* (6.85%), *F. acuminatum* (5.48%), *F. glycines* (5.48%), *F. temperatum* (2.74%), *F. armeniacum* (2.74%), *Fusarium* sp. (2.05%), *F. flagelliforme* (1.4%), and *F. annulatum* (0.68%). The *Fusarium incarnatum-equiseti* species complex (FIESC, including *F. ipomoeae*, *F. compactum*, *F. citri*, and *F. flagelliforme*) was the most prevalent, indicating an evolving occurrence of the Fusarium species causing maize leaf blight. The typical symptoms observed on the maize leaves were oval to long strip lesions, with a gray to dark gray or brownish red coloration in the center and a chlorotic area at the edges. Based on the *tef1* gene, seven haplotypes of FIESC were identified in Heilongjiang province, suggesting a population expansion. This is the first report of *F. ipomoeae*, *F. compactum*, *F. flagelliforme*, *F. citri*, *F. sporotrichioides*, *F. graminearum*, *F. asiaticum*, *F. acuminatum*, *F. glycines*, *F. temperatum*, *F. armeniacum, Fusarium* sp., and *F. annulatum* causing maize leaf blight in Heilongjiang province, China. The current research is informative for managing disease, exploring the phylogenetic relationship among *Fusarium* species, and clarifying the diversity of Fusarium species associated with maize leaf blight.

## 1. Introduction

*Fusarium* spp. can cause several diseases in maize, such as Fusarium ear rot [1,2,3], Fusarium stalk rot and root rot [2,4], seedling blight [5], and maize leaf blight [6]. Regarding maize leaf blight, Fusarium verticillioides was the first pathogen, reported in 1968 [6], to cause the disease, and the only reported one up to now. However, the pathogenicity and diversity of *Fusarium* spp. causing maize leaf blight are still unclarified. Maize leaf blight is characterized by symptoms of irregular or spindle lesions, with gray to reddish brown coloration in the lesions’ center surrounded by a chlorotic halo. Sometimes, this disease is misjudged as northern corn leaf spot due to the similar symptoms in the field. Thus, the identification of the pathogens based only on disease symptoms in the field is difficult.

To our knowledge, the genus *Fusarium* includes more than 300 phylogenetic species [7] and is one of the most important plant pathogens in the world [8]. Most species within the genus can produce a diverse range of mycotoxins, causing varying degrees of acute or chronic toxic effects [1]. Therefore, the accurate identification of these mycotoxin producers is a considerable endeavor [9]. For the identification of fungi and the investigation of molecular ecology, the internal transcribed spacer (ITS) is the most sequenced DNA region [10]. However, the ITS region cannot distinguish the species complex of *Fusarium* due to its conservation [11]. By contrast, the *tef1* gene can be used to discriminate *Fusarium* species at the species or subspecies level [11,12], and the *rpb2* gene is also more informative and frequently employed, so it has been recommended that they are sequenced for *Fusarium* species identification. However, although the partial beta-tubulin gene has been used to identify several *Fusarium* species, it was not universally informative within *Fusarium* [13].

The members of *Fusarium incarnatum*-*equiseti* species complex (FIESC) are considered important plant pathogens. FIESC is rarely considered the major pathogen of disease epidemics, but it has been identified as a co-occurring fungal pathogen during an infection [14]. Thirty phylogenetic species within the FIESC (FIESC 1 through FIESC 30) were recognized through Multi-locus Sequence Typing (MLST) [15,16], and the species containing multiple haplotypes are designated by the addition of a lowercase letter to the phylogenetic species designation [9].

Phylogenetic and genetic diversity analyses based on multiple sequences can reveal evolutionary relationships associated with geographical regions [9]. High genetic diversity indicates greater adaptability to changing environmental conditions. In some complex evolutionary scenarios, appropriate and sufficient information may not be obtained from phylogenetic trees [17,18]. By comparison, haplotype networks can be employed to analyze the intraspecific diversity of populations, genetic processes, and the biogeography and history of populations [18,19].

To date, there has been little research on pathogenicity, genetic diversity, and the haplotype groups of pathogenic *Fusarium* species isolated from symptomatic maize leaves in China. Hence, the purposes of the present study were to: (i) describe the morphological characterization and phylogenetic relationships based on *tef1* and *rpb2* genes of *Fusarium* species responsible for maize leaf blight in Heilongjiang province, (ii) evaluate the pathogenicity of different *Fusarium* species, and (iii) determine the haplotype diversity of FIESC based on *tef1* associated with maize leaf blight.

## 2. Materials and Methods

### 2.1. Fusarium Isolates Collection

From 2019 to 2021, a total of 132 symptomatic maize leaves were collected from 10 different maize-growing counties or cities in Heilongjiang province. The symptomatic maize leaves were cut with a sterilized scalpel, superficially disinfected with a 2% solution of sodium hypochlorite for 1 min and 75% ethanol for 30 s, rinsed thrice with sterile distilled water, and air-dried on sterile filter papers under aseptic conditions. Pure cultures were obtained by single-spore isolation and maintained on PDA (potato dextrose agar) at 25 °C for 7 days. *Fusarium* isolates were obtained and preserved on PDA slants at 4 °C and 20% glycerol at −80 °C for temporary storage and long-term storage, respectively.

### 2.2. Morphological Characterization

All *Fusarium* isolates were incubated on PDA plate in the dark at 25 °C for 7 days. Colony color and colony texture were observed for each isolate. To determine the size of well-developed macroconidia (*n* = 30) and the number of septa, these *Fusarium* isolates were incubated on PDA plates at 25°C for 7 days with light/dark cycle of 8/16 h. The macroconidia were observed under light microscopy (Zeiss Axiolab5 equipped with an Axiocam 208 color industrial digital camera).

### 2.3. DNA Extraction and Sequence Analysis

Fresh mycelia were harvested from cultures grown on PDA supplemented with streptomycin (50 mg/L) and tetracycline (50 mg/L) for 7 days at 28 °C. The extraction of fungal genomic DNA was performed as Ramdial et al. described [9]. The sequences of the translation elongation factor 1-alpha (*tef1*) gene, second-largest subunit of RNA polymerase II gene (*rpb2*), and partial beta-tubulin gene were amplified by the primers EF-1/EF-2, RPB2-5f2/RPB2-7cr, and Bt2a/Bt2b [13,20], respectively. The PCR products were sent to Jilin Comate Bioscience Co. Ltd. for purification and sequencing. Sequences of 146 Fusarium isolates were searched against GenBank and FUSARIOID-ID database (www.fusarium.org, accessed date: 1 September 2022) [21] by Basic Local Alignment Search Tool (BLAST) analysis and then deposited into the NCBI GenBank (Table 1).

### 2.4. Phylogenetic Relationships among Fusarium Isolates

The *rpb2* (794–896 bp), *tef1*(546–686 bp), and β-tubulin (332–356 bp) gene sequences of *Fusarium* isolates were also compared to the sequences available in the FUSARIOID-ID database (www.fusarium.org, accessed date: 1 September 2022) to collect related sequences for inclusion in phylogenetic analysis. Multiple sequence alignments were correspondingly inferred in Molecular Evolutionary Genetics Analysis (MEGA) 7 software [22] using the MUSCLE (multiple sequence comparison by log-expectation) program [23] and refined manually if necessary. To generate concatenated datasets, single gene sequences (*tef1* and *rpb2*) were manually combined utilizing BioEdit [24]. Phylogenetic tree based on the concatenated sequences of *tef1* and *rpb2* genes was built using the maximum likelihood (ML) method in MEGA 7, respectively. ML tree was generated from bootstrapping 1000 replicates. Bootstrap values ≥ 70% were shown in phylogenetic trees. The sequences from the *Fusarium* spp. type strains, initially identified as closely related to the sequences herein, were finally included by the preliminary BLAST searches.

### 2.5. Pathogenicity Tests

All *Fusarium* isolates were used to evaluate their pathogenicity based on the method described by Xu et al. [25]. To fulfill Koch’s postulates, 10 healthy, surface-sterilized, and four to five leaf-stage maize seedlings (var. Demeiya 3) for each *Fusarium* isolate were inoculated with *Fusarium* spore suspension (1 × 10^6^ spores/mL). Twenty maize seedlings sprayed with sterile distilled water served as controls. All seedlings sealed with plastic bags were maintained in a greenhouse at 25 °C with 90% relative humidity and a light/dark cycle of 12/12 h.

Disease severity (DS) and disease incidence (DI) were assessed 14 days post-inoculation. DS was measured based on a 0–9 scale described by Rafael et al. [26] and Xu et al. [25]: 0 (no visible symptoms), 1 (0 up to 0.5%), 2 (0.5–1.6%), 3 (1.6–5.0%), 4 (5.0–15%), 5 (15–37%), 6 (37–66%), 7 (66–87%), 8 (87% to 96%), and 9 (96–100%). DI was computed by following formula: DI = [100 × ∑ (*n* × corresponding DS)]/(N × 9), where *n* is the number of infected inoculation leaves corresponding to each disease rating, and N is the total number of inoculation leaves. Disease incidence was computed by following formula: disease incidence = number of diseased leaves/total number of inoculated leaves of living maize plants. A least significant difference (LSD) test was used for statistical analysis at a significance level of *p* < 0.05 with the Statistical Package for Social Sciences (SPSS) software (v. 20.0; SPSS Inc., Wacker Drive, Chicago, IL, USA, Illinois.IBM Corp., 2012. IBM). All re-isolated pathogens from inoculated maize leaves were identified using morphological and molecular methods mentioned above. Each experiment was repeated two times.

### 2.6. DNA Polymorphism

DNA Sequence Polymorphism software version 6 was used to individually determine the DNA polymorphism relative degree of the *tef1* gene sequences [27]. Furthermore, Tajima’s D, Fu and Li’s D, and Fu and Li’s F were used to determine neutrality test statistics. Significant values of these tests indicate the presence of population changes [28,29]. DNA polymorphism analyses were only performed on FIESC and not on other *Fusarium* species on account of the limited number of isolates from those species obtained in the current study.

### 2.7. Haplotype Analysis

Haplotype networks were individually generated based on the *tef1* gene sequences of 70 FIESC isolates (including 30 *F. ipomoeae* isolates, 25 *F. compactum* isolates, 13 *F. citri* isolates, and 2 *F. flagelliforme* isolates in the present study) using PopART v. 1.7 (Allan Wilson Centre Imaging Evolution Initiative) to evaluate genealogy pattens of the haplotypes [19]. The aligned haplotype sequences were used to construct a TCS network [30,31].

## 3. Results

### 3.1. Fungal Isolation and Morphological Characterization

In this study, 146 *Fusarium* isolates were obtained from symptomatic maize leaves in China (Table 1), which were initially classified into 11 groups based on their morphological features, including the *Fusarium incarnatum-equiseti* species complex (FIESC, including *F. ipomoeae, F. compactum, F. citri,* and *F. flagelliforme* in this study), *F. sporotrichioides, F. armeniacum, F. asiaticum, F. graminearum, Fusarium sp., F. acuminatum, F. glycines, F. annulatum, F. temperatum*, and *F. verticillioides* (Table 2).

Seventy isolates were identified as the members of FIESC and produced white to light yellow aerial mycelia. The bottom of the plate turned white to pale brown with time. The macroconidia were slightly curved at the apex with three to five septa and ranged from 39.6 to 83.5 × 3.9 to 5.2 μm (*n* = 30, Figure 1a–d and Figure 2a–d) in size.

Fourteen *F. sporotrichioides* isolates produced dense, pinkish white to carmine red aerial mycelia, whose macroconidia were moderately curved to straight with three to five septa, but mostly three-septate, and measured 20.5 to 47.3µm × 2.8 to 4.2 µm (*n* = 30, Figure 1n and Figure 2f).

The colonies of four *F. armeniacum* isolates were white to light pink. The macroconidia were prominently curved with three to five septa and had sizes ranging from 35.6 to 59.3 μm × 4 to 4.6 μm (*n* = 30, Figure 1g and Figure 2g).

Ten isolates producing pink to fluffy dark red aerial mycelia, and red to aubergine pigmentation with age, were classified under *F. asiaticum*. Their macroconidia were falcate with three to five septa and measured 25.2 to 61.5 × 3.9 to 4.7 μm (*n* = 30, Figure 1h and Figure 2h).

Fourteen *F. graminearum* isolates produced white-pink aerial mycelia and had dark red pigmentation. Their macroconidia were straight or slightly curved with five to seven septa and measured 25.4 to 97.7 × 3.4 to 5.8 µm (*n* = 30, Figure 1k and Figure 2i).

Three *Fusarium* sp. isolates produced white to yellow colonies and red pigmentation. Their macroconidia were curved with three to five septa and measured 34.0 to 71.6 × 3.2 to 4.7 μm (*n* = 30, Figure 1j and Figure 2j).

The colonies of eight *F. acuminatum* isolates were whitish-pink or carmine to rose red. Their macroconidia were slender with a distinct curve of the apical cell, mostly three- to five-septate, and measured 31.3 to 65.3 × 4.0 to 6.5 µm (*n* = 30, Figure 1f and Figure 2k).

The colonies of eight *F. glycines* isolates produced fluffy, white aerial hyphae and a dark red pigment. Their macroconidia were three- to seven-septate, slightly curved, and ranged from 53.3 to 117.9 μm × 3.3 to 4.5 μm (*n* = 30, Figure 1l and Figure 2l) in size.

The aerial mycelia of the *F. annulatum* isolates were white to cream-colored and turned violet with age, and their macroconidia were straight or slightly curved and contained three to five septa, with sizes of 21.5 to 58.3 × 2.1 to 3.6 µm (*n* = 30, Figure 1i and Figure 2m).

The colonies of four *F. temperatum* isolates were pinkish-white and produced mostly three-septate macroconidia. Their macroconidia measured 34.5 to 60.8 × 3.2 to 4.1 µm (*n* = 30, Figure 1e and Figure 2n).

Ten *F. verticillioides* isolates formed cottony white to greyish-purple colonies with a dark yellow to purple-gray underside. Their microconidia were abundant and mainly showed clavate shapes measuring 4.2 to 7.5 × 2.1 to 3.8 μm (*n* = 30, Figure 1m and Figure 2e). However, there were no macroconidia of the F. verticillioides isolates observed in this study.

### 3.2. Phylogenetic Analysis

The sequences of the *tef1*, *rpb2,* and beta-tubulin genes of all the *Fusarium* isolates obtained in this study were searched against the FUSARIOID-ID database (www.fusarium.org, accessed date: 1 September 2022) using a BLAST analysis (Appendix A). For further molecular verification, a multilocus phylogenetic analysis (MLSA) was further performed based on the concatenated sequences (tef1 and rpb2 genes) of all the *Fusarium isolates* (Figure 3). These results indicated that all the Fusarium isolates could be grouped into 14 clades, including F. *ipomoeae, F. compactum, F. sporotrichioides, F. citri, F. graminearum, F. asiaticum, F. verticillioides, F. acuminatum, F.glycines, F. temperatum, F. armeniacum, Fusarium* sp., *F. flagelliforme*, and *F. annulatum*.

### 3.3. Pathogenicity Tests

Two weeks after inoculation, the pathogenicity test revealed that all the Fusarium species could cause similar maize leaf blight symptoms (Figure 4). Small oval to fusiform or long striped spots initially appeared on the maize leaves three days post-inoculation, in which the lesions’ centers were gray to reddish brown and surrounded by a chlorotic area. The lesions gradually enlarged with time and merged into each other. In a severe case, the infected leaves were withered. The symptoms observed under greenhouse conditions were similar to the symptoms of maize leaf blight in the field (Figure 4a). No symptoms were observed in the control group. In addition, all the *Fusarium* species were consistently re-isolated and confirmed based on morphological and molecular methods, while no *Fusarium* isolates were obtained from the control group, thus fulfilling Koch’s postulates. The average disease incidence and average disease index caused by the *Fusarium* species ranged from 23 to 74% and from 52 to 85, respectively (Figure 5 and Figure 6; Appendix A). Moreover, all the *Fusarium* isolates were pathogenic towards maize leaves (var. Demeiya 3) and caused maize leaf blight in the inoculation study. In addition, *F. graminearum* showed the highest virulence, followed by *Fusarium* sp., *F. glycines, F. acuminatum, F. compactum, F. temperatum, F. asiaticum, F. citri, F. verticillioides, F. armeniacum, F. ipomoeae, F. annulatum, F. sporotrichioides*, and *F. flagelliforme.*

### 3.4. Haplotype Analyses and DNA Polymorphism

The haplotype networks based on the tef1 gene sequences of 70 FIESC isolates (including 30 *F. ipomoeae* isolates, 25 *F. compactum* isolates, 2 *F. flagelliforme* isolates, and 13 *F. citri* isolates) obtained in this study were used to determine evolutionary relationships among the haplotypes. Most haplotypes within one species were closely related and separated by one to three mutations.

A total of seven haplotypes were identified: the *F. ipomoeae* isolates were assigned to Hap 1 and 4; *F. compactum* isolates were assigned to Hap 2, 5, and 6; *F. flagelliforme* isolates were assigned to Hap 3; and *F. citri* isolates were assigned to Hap 7 (Figure 7).

Meanwhile, Hap 1, 2, 4, 5, and 7 were shared haplotypes (Figure 7). Hap 1 was the most predominant haplotype, and presented in six locations (Harbin city, Wuchang city, Daqing city, Suihua city, Jixi city, and Qiqihar city). Hap 2 was found in Harbin city and Jixi city. Hap 4 was found in Harbin city and Wuchang city. Hap 5 was distributed in Harbin city and Shuangyashan city. Hap 7 was detected in Harbin city and Qitaihe city. Furthermore, two private haplotypes (Hap 3 and 6) were present in Harbin city and Jixi city, respectively. However, there was no obvious center between these predominant haplotypes. In addition, A low degree of nucleotide diversity (0.02706) and a high degree of haplotype diversity (Hd) (0.778) were found. Tajima’s D, Fu and Li’s D, and Fu and Li’s F tests were negative with no significance (*p* > 0.10, Appendix A).

## 4. Discussion

As far as we know, this is the first systematic study of the *Fusarium* species associated with maize leaf blight. In this study, 146 *Fusarium* isolates delimited to 14 *Fusarium* species were obtained from symptomatic maize leaves in Heilongjiang province. To analyze the genetic relationship between these *Fusarium* isolates obtained in the current study, phylogenetic trees were constructed only based on the concatenated sequences of *tef1* and *rpb2* genes because these two genes were more informative and frequently employed, while the beta-tubulin gene was not universally informative in *Fusarium* [13]. A total of 14 *Fusarium* species were identified, including *F. ipomoeae*, *F. compactum*, *F. sporotrichioides*, *F. citri*, *F. graminearum*, *F. asiaticum*, *F. verticillioides*, *F. acuminatum*, *F. glycines*, *F. temperatum*, *F. armeniacum*, *Fusarium* sp., *F. flagelliforme*, and *F. annulatum*. Except for *F. verticillioides*, which was the only reported pathogen inciting maize leaf blight [6], the remaining *Fusarium* species were all first reported in Heilongjiang province, China, suggesting that the composition of *Fusarium* species causing maize leaf blight may have changed.

Furthermore, considerable pathogenicity differences were found among the different *Fusarium* species. *F. graminearum* showed significantly greater average disease incidence and average disease indices than those of other *Fusarium* species, followed by *Fusarium* sp., *F. glycines*, *F. acuminatum*, *F. compactum*, *F. temperatum*, *F. asiaticum*, *F. citri*, *F. verticillioides*, *F. armeniacum*, *F. ipomoeae*, *F. annulatum*, *F. sporotrichioides*, and *F. flagelliforme*. Members of FIESC are generally considered co-occurring pathogens [32,33], and the moderate aggressiveness of FIESC in this study seems to confirm the previous conclusion. FIESC was the most predominant in this study. Members of FIESC have been frequently isolated from maize, soybean, rice, barley, wheat, and so on [34,35,36,37,38,39] and have also been reported to cause leaf blight in peanut plants [40] and *Cyperus iria* [41].

The haplotype groups of FIESC associated with maize leaf blight were first identified in this work. The predominant haplotype (Hap 1) represented multiple locations (Harbin city, Wuchang city, Daqing city, Suihua city, Jixi city, and Qiqihar city). It is well-known that older haplotypes may have a wider geographic distribution, which suggests that Hap 1 has lasted in the population for a long time [42]. The rest of the haplotypes may represent recently evolved lineages [4]. Furthermore, haplotypes 2, 5, and 6 belonged to the *F. compactum* clade; haplotypes 1 and 4 belonged to the *F. ipomoeae* clade; haplotype 3 belonged to the *F. flagelliforme* clade; and haplotype 7 belonged to the *F. citri* clade. These FIESC isolates were distributed in different clades in the haplotype network, which suggests that the haplotype network could effectively differentiate the *Fusarium* species complex and further confirmed our identification results. Moreover, the *F. flagelliforme* haplotype (Hap 3) and *F. citri* haplotype (Hap 7) were observed in external parts of the haplotype network and showed more mutation events from their nearest haplotypes, which indicated that these two species have an older evolutionary relationship. In addition, the high haplotype diversity and low nucleotide diversity indicated a population expansion [43].

In conclusion, the current study focused on the pathogenicity and genetic diversity of *Fusarium* species causing maize leaf blight in Heilongjiang province, China, and is the first to report *F. ipomoeae, F. compactum, F. flagelliforme*, *F. citri*, *F. sporotrichioides*, *F. graminearum*, *F. asiaticum*, *F. verticillioides*, *F. acuminatum*, *F. glycines*, *F. temperatum*, *F. armeniacum*, *Fusarium* sp., and *F. annulatum* as the causal agents. *Fusarium* can cause various maize diseases; therefore, clarifying the population composition of *Fusarium* spp. on maize leaves will provide information for the overall control of maize diseases.

## Figures and Tables

**Figure 1 jof-08-01170-f001:**
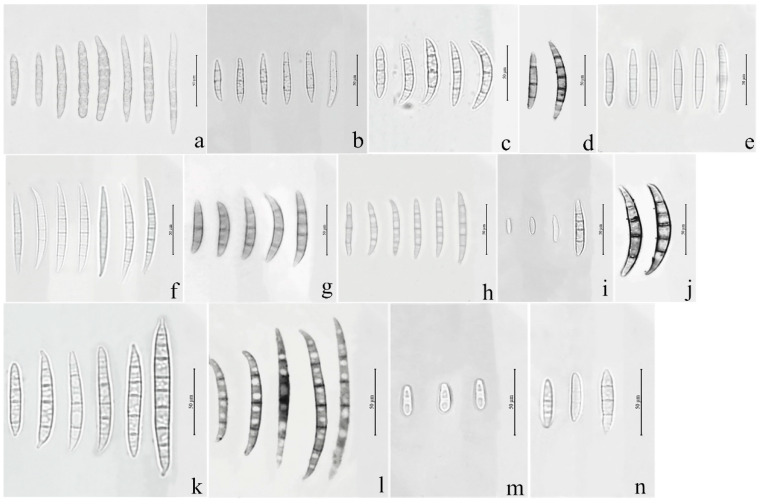
Macroconidia or microconidia of representative isolates of 14 *Fusarium* species. (**a**) *F. compactum*; (**b**) *F. ipomoeae*; (**c**) *F. citri*; (**d**) *F. flagelliforme*; (**e**) *F. temperatum*; (**f**) *F. acuminatum*; (**g**) *F. armeniacum*; (**h**) *F. asiaticum*; (**i**) *F. annulatum*; (**j**) *Fusarium* sp.; (**k**); *F. graminearum*; (**l**) *F. glycines*; (**m**) *F. verticillioides*; (**n**) *F. sporotrichioides*.

**Figure 2 jof-08-01170-f002:**
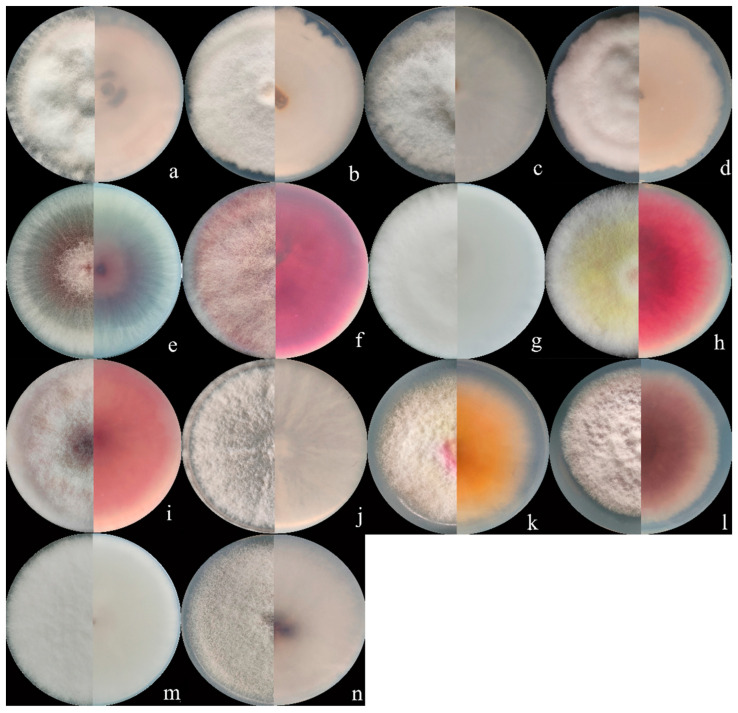
Colony appearance of representative isolates of 14 *Fusarium* species. (**a**) *F. compactum*; (**b**) *F. ipomoeae*; (**c**) *F. citri*; (**d**) *F. flagelliforme*; (**e**) *F. verticillioides* (**f**) *F. sporotrichioides*; (**g**) *F. armeniacum*; (**h***) F. asiaticum*; (**i**) *F. graminearum*; (**j**) *Fusarium* sp.; (**k**) *F. acuminatum*; (**l**) *F. glycines*; (**m**) *F. annulatum*; (**n**) *F. temperatum*.

**Figure 3 jof-08-01170-f003:**
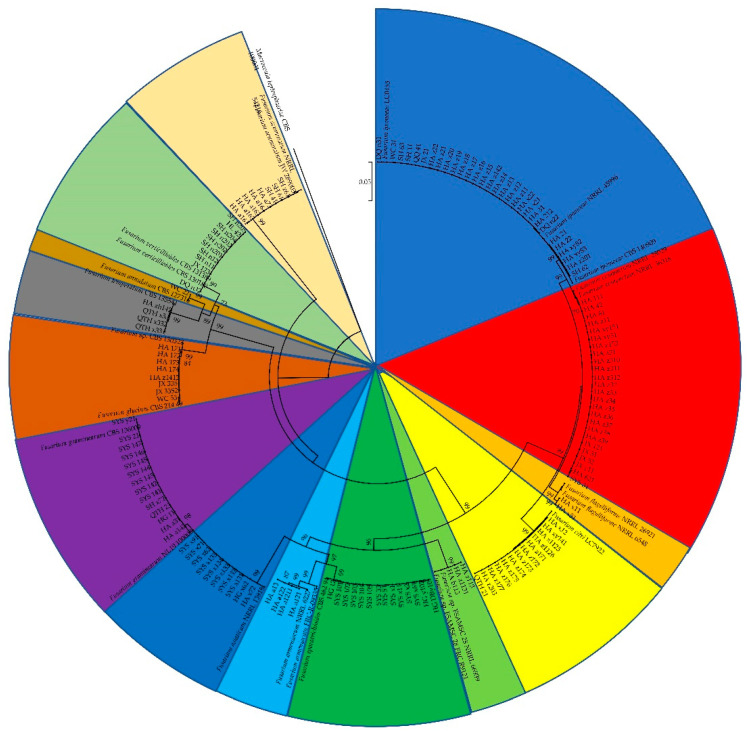
Phylogenetic tree obtained from maximum likelihood analysis based on the concatenated sequences of *tef1* and *rpb2* genes. Support values at nodes representing RA × ML bootstrap percentages with values ≥70 are shown above the branches.

**Figure 4 jof-08-01170-f004:**
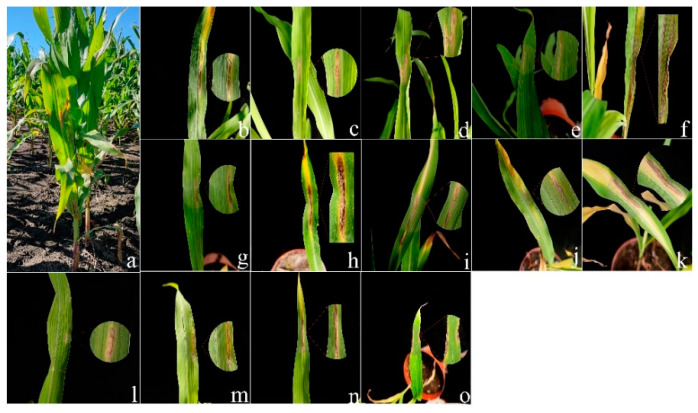
(**a**) Leaf blight symptoms on maize leaves caused by *Fusarium* species in the field; (**b**–**o**) Typical symptoms observed in greenhouse on maize leaves after inoculation with: (**b**) *F. ipomoeae*; (**c**) *F. compactum*; (**d**) *F. flagelliforme*; (**e**) *F. asiaticum*; (**f**) *F. armeniacum*; (**g**) *F. citri*; (**h**) *F. sporotrichioides*; (**i**) *Fusarium* sp.; (**j**) *F. glycines*; (**k**) *F. graminearum*; (**l**) *F. annulatum*; (**m**) *F. temperatum*; (**n**) *F. verticillioides*; (**o**) *F. acuminatum*.

**Figure 5 jof-08-01170-f005:**
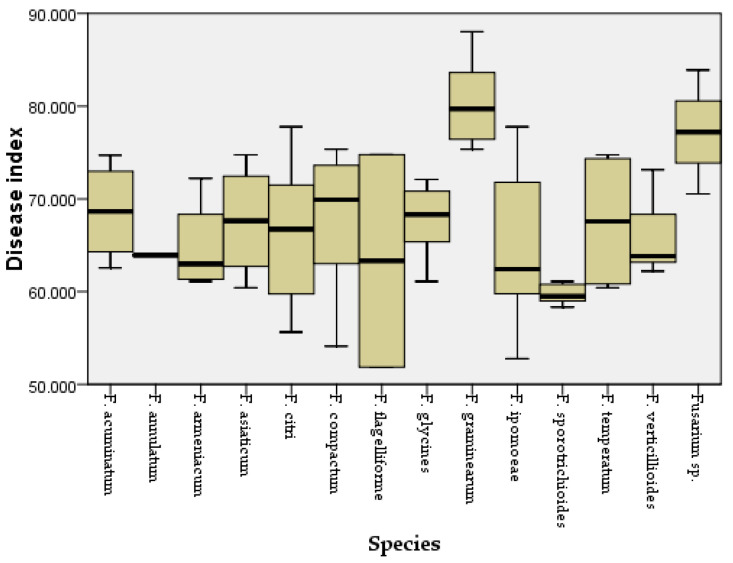
Disease index for maize leaves inoculated with different *Fusarium* species.

**Figure 6 jof-08-01170-f006:**
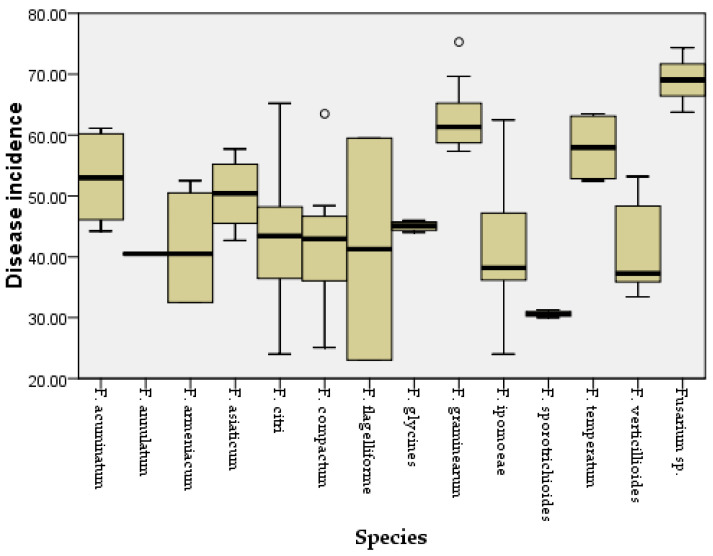
Disease incidence for maize leaves inoculated with different *Fusarium* species. Outliers are represented by a hollow circle.

**Figure 7 jof-08-01170-f007:**
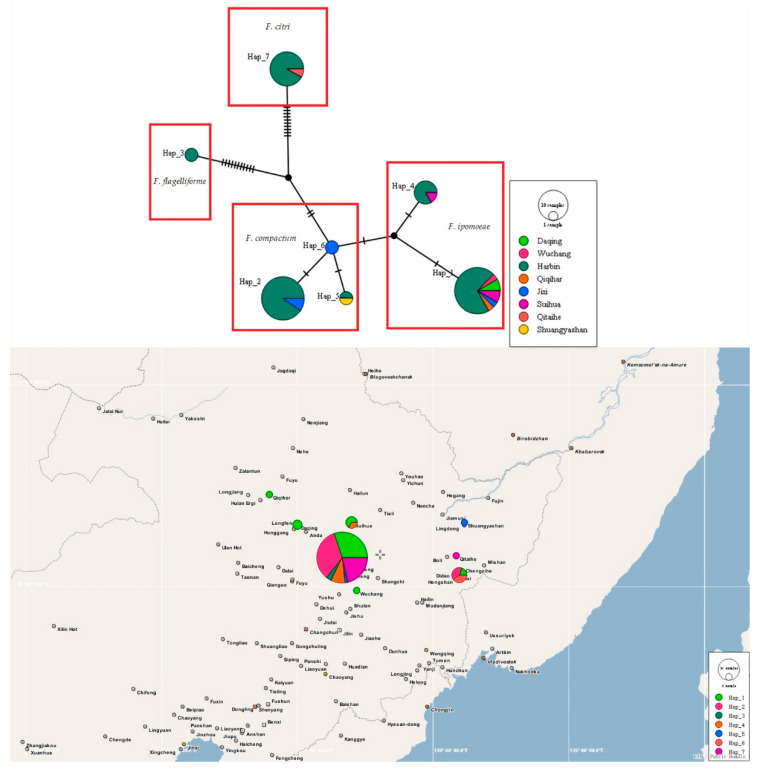
TCS analyses and the haplotype distribution based on the *tef1* gene sequences of 70 FIESC isolates obtained in this study. Each haplotype is represented by a circle, the size of which is proportional to the haplotype frequency.

**Table 1 jof-08-01170-t001:** List of GenBank accession numbers of Fusarium isolates obtained from symptomatic maize leaves collected from Heilongjiang province and reference strains used in this study.

Isolates.	Latitude and Longitude	Species	GenBank Accession Nos.		
			*tef1*	*rpb2*	Beta-Tubulin
HA-z142	126.738196, 45.753014	*F. ipomoeae*	OM985077	OP436018	OP642121
HA-z11	126.738196, 45.753014	*F. ipomoeae*	OM985078	OP436019	OP642120
HA-z12	126.738196, 45.753014	*F. ipomoeae*	OM985079	OP436020	OP642119
HA-z13	126.738196, 45.753014	*F. ipomoeae*	OM985080	OP436021	OP642118
HA-z14	126.738196, 45.753014	*F. ipomoeae*	OM985081	OP436022	OP642117
HA-z15	126.738196, 45.753014	*F. ipomoeae*	OM985082	OP436023	OP642116
HA-z16	126.738196, 45.753014	*F. ipomoeae*	OM985083	OP436024	OP642115
HA-z17	126.738196, 45.753014	*F. ipomoeae*	OM985084	OP436025	OP642114
HA-z18	126.738196, 45.753014	*F. ipomoeae*	OM985085	OP436026	OP642113
HA-z19	126.738196, 45.753014	*F. ipomoeae*	OM985086	OP436027	OP642112
HA-z20	126.738196, 45.753014	*F. ipomoeae*	OM985087	OP436028	OP642111
HA-z21	126.738196, 45.753014	*F. ipomoeae*	OM985088	OP436029	OP642110
HA-z22	126.738196, 45.753014	*F. ipomoeae*	OM985089	OP436030	OP642109
HA-x22	126.868024, 45.850128	*F. ipomoeae*	OM985106	OP436031	OP642108
HA-xy82	126.933932, 45.769353	*F. ipomoeae*	OM985109	OP436032	OP642122
HA-xy83	126.933932, 45.769353	*F. ipomoeae*	OM985110	OP436033	OP642123
HA-31	126.868024, 45.850128	*F. ipomoeae*	OM985118	OP436034	OP642107
SH-11	127.270457, 46.64457	*F. ipomoeae*	OM985119	OP436035	OP642106
SH-63	127.270457, 46.64457	*F. ipomoeae*	OM985120	OP436036	OP642105
WC-31	127.22506, 44.93996	*F. ipomoeae*	OM985124	OP436037	OP642104
QQ-41	124.340195, 47.29158	*F. ipomoeae*	OM985125	OP436038	OP642103
SH-62	127.270457, 46.64457	*F. ipomoeae*	OM985126	OP436039	OP642124
HA-z201	126.738196, 45.753014	*F. ipomoeae*	OM985127	OP436040	OP642125
HA-21	126.868024, 45.850128	*F. ipomoeae*	OM985128	OP436041	OP642126
HA-22	126.868024, 45.850128	*F. ipomoeae*	OM985129	OP436042	OP642127
HA-x21	126.868024, 45.850128	*F. ipomoeae*	OM985130	OP436043	OP642102
HA-212	126.868024, 45.850128	*F. ipomoeae*	OM985140	OP436044	OP642101
DQ-n22	125.835845, 46.329205	*F. ipomoeae*	OM985182	OP436045	OP642100
JX-21	132.477436, 46.339951	*F. ipomoeae*	OM985183	OP436046	OP642098
DQ-n31	125.835845, 46.329205	*F. ipomoeae*	OM985184	OP436047	OP642099
HA-61	126.868024, 45.850128	*F. compactum*	OM985144	OP435951	OP642130
HA-111	126.868024, 45.850128	*F. compactum*	OM985102	OP435952	OP642131
JX-y11	132.477436, 46.339951	*F. compactum*	OM985123	OP435953	OP642132
HA-621	126.868024, 45.850128	*F. compactum*	OM985145	OP435975	OP642128
SYS-31	132.768479, 46.215238	*F. compactum*	OM985146	OP435954	OP642129
HA-z152	126.738196, 45.753014	*F. compactum*	OM985147	OP435955	OP642133
HA-z31	126.738196, 45.753014	*F. compactum*	OM985148	OP435956	OP642134
HA-z32	126.738196, 45.753014	*F. compactum*	OM985149	OP435957	OP642135
HA-z33	126.738196, 45.753014	*F. compactum*	OM985150	OP435958	OP642136
HA-z34	126.738196, 45.753014	*F. compactum*	OM985151	OP435959	OP642137
HA-z35	126.738196, 45.753014	*F. compactum*	OM985152	OP435960	OP642138
HA-z36	126.738196, 45.753014	*F. compactum*	OM985153	OP435961	OP642139
HA-z37	126.738196, 45.753014	*F. compactum*	OM985154	OP435962	OP642140
HA-z38	126.738196, 45.753014	*F. compactum*	OM985155	OP435963	OP642141
HA-z39	126.738196, 45.753014	*F. compactum*	OM985156	OP435964	OP642142
HA-z310	126.738196, 45.753014	*F. compactum*	OM985157	OP435965	OP642143
HA-z311	126.738196, 45.753014	*F. compactum*	OM985158	OP435966	OP642144
HA-z312	126.738196, 45.753014	*F. compactum*	OM985159	OP435967	OP642145
HA-xy151	126.933932, 45.769353	*F. compactum*	OM985160	OP435968	OP642146
HA-xy31	126.933932, 45.769353	*F. compactum*	OM985161	OP435969	OP642147
HA-a11	126.868024, 45.850128	*F. compactum*	OM985162	OP435970	OP642152
HA-42	126.868024, 45.850128	*F. compactum*	OM985163	OP435971	OP642148
JX-52	132.477436, 46.339951	*F. compactum*	OM985164	OP435972	OP642149
JX-121	132.477436, 46.339951	*F. compactum*	OM985165	OP435973	OP642150
JX-31	132.477436, 46.339951	*F. compactum*	OM985166	OP435974	OP642151
HA-x12	126.868024, 45.850128	*F. citri*	OM985167	OP435950	OP642166
QTH-21	131.139405, 45.733699	*F. citri*	OM985168	OP435949	OP642167
HA-z1125	126.738196, 45.753014	*F. citri*	OM985169	OP435948	OP642158
HA-z171	126.738196, 45.753014	*F. citri*	OM985170	OP435947	OP642165
HA-z172	126.738196, 45.753014	*F. citri*	OM985171	OP435946	OP642164
HA-z173	126.738196, 45.753014	*F. citri*	OM985172	OP435945	OP642163
HA-z174	126.738196, 45.753014	*F. citri*	OM985173	OP435944	OP642162
HA-z175	126.738196, 45.753014	*F. citri*	OM985174	OP435943	OP642161
HA-z176	126.738196, 45.753014	*F. citri*	OM985175	OP435942	OP642160
HA-z177	126.738196, 45.753014	*F. citri*	OM985176	OP435941	OP642159
HA-z1126	126.738196, 45.753014	*F. citri*	OM985177	OP435940	OP642157
HA-xy141	126.933932, 45.769353	*F. citri*	OM985178	OP435939	OP642156
HA-z203	126.738196, 45.753014	*F. citri*	OM985179	OP435938	OP642155
HA-x11	126.868024, 45.850128	*F. flagelliforme*	OM985104	OP435921	OP642153
HA-x51	126.868024, 45.850128	*F. flagelliforme*	OM985105	OP435920	OP642154
HA-a31	126.868024, 45.850128	*F. graminearum*	OM985090	OP435980	OP642200
HG-11	130.440826, 47.312952	*F. graminearum*	OM985091	OP435981	OP642201
QTH-23	131.139405, 45.733699	*F. graminearum*	OM985103	OP435982	OP642202
SH-x72	127.270457, 46.64457	*F. graminearum*	OM985108	OP435983	OP642203
SYS-y21	132.768479, 46.215238	*F. graminearum*	OM985111	OP435984	OP642204
SYS-21	132.768479, 46.215238	*F. graminearum*	OM985199	OP435985	OP642205
SYS-141	132.768479, 46.215238	*F. graminearum*	OM985200	OP435986	OP642206
SYS-142	132.768479, 46.215238	*F. graminearum*	OM985201	OP435987	OP642207
SYS-143	132.768479, 46.215238	*F. graminearum*	OM985202	OP435988	OP642208
SYS-144	132.768479, 46.215238	*F. graminearum*	OM985203	OP435989	OP642209
SYS-145	132.768479, 46.215238	*F. graminearum*	OM985204	OP435990	OP642210
SYS-146	132.768479, 46.215238	*F. graminearum*	OM985205	OP435991	OP642211
SYS-147	132.768479, 46.215238	*F. graminearum*	OM985206	OP435992	OP642212
HA-a142	126.868024, 45.850128	*F. graminearum*	OM985138	OP435993	OP642213
SYS-x71	131.583118, 46.462499	*F. asiaticum*	OM985092	OP436053	OP642088
SYS-x91	131.583118, 46.462499	*F. asiaticum*	OM985093	OP436054	OP642089
HA-x72	126.868024, 45.850128	*F. asiaticum*	OM985094	OP436055	OP642090
HG-x62	130.440826, 47.312952	*F. asiaticum*	OM985095	OP436056	OP642091
SYS-x62	131.583118, 46.462499	*F. asiaticum*	OM985096	OP436057	OP642092
SYS-x131	131.583118, 46.462499	*F. asiaticum*	OM985097	OP436058	OP642093
SYS-x132	131.583118, 46.462499	*F. asiaticum*	OM985098	OP436059	OP642094
SYS-x133	131.583118, 46.462499	*F. asiaticum*	OM985099	OP436060	OP642095
SYS-x134	131.583118, 46.462499	*F. asiaticum*	OM985100	OP436061	OP642096
SYS-x135	131.583118, 46.462499	*F. asiaticum*	OM985101	OP436062	OP642097
HA-zh142	126.738196, 45.753014	*F. temperatum*	OM985107	OP436049	OP642174
QTH-X332	131.139405, 45.733699	*F. temperatum*	OM985131	OP436050	OP642171
QTH-X331	131.139405, 45.733699	*F. temperatum*	OM985132	OP436051	OP642173
QTH-X33	131.139405, 45.733699	*F. temperatum*	OM985133	OP436052	OP642172
HA-z113	126.738196, 45.753014	*Fusarium* sp.	OM985112	OP436063	OP642168
HA-b113	126.738196, 45.753014	*Fusarium* sp.	OM985113	OP436064	OP642170
HA-Z1131	126.738196, 45.753014	*Fusarium* sp.	OM985143	OP436065	OP642169
SYS-x11	131.583118, 46.462499	*F. sporotrichioides*	OM985209	OP436017	OP642176
SYS-x61	131.583118, 46.462499	*F. sporotrichioides*	OM985210	OP436016	OP642177
SYS-x1	131.583118, 46.462499	*F. sporotrichioides*	OM985211	OP436015	OP642178
SYS-x2	131.583118, 46.462499	*F. sporotrichioides*	OM985212	OP436014	OP642179
HG-12	130.440826, 47.312952	*F. sporotrichioides*	OM985213	OP436013	OP642180
SYS-33	132.768479, 46.215238	*F. sporotrichioides*	OM985214	OP436012	OP642181
SYS-101	132.768479, 46.215238	*F. sporotrichioides*	OM985215	OP436011	OP642182
SYS-102	132.768479, 46.215238	*F. sporotrichioides*	OM985216	OP436010	OP642183
SYS-103	132.768479, 46.215238	*F. sporotrichioides*	OM985217	OP436009	OP642184
SYS-104	132.768479, 46.215238	*F. sporotrichioides*	OM985218	OP436008	OP642185
SYS-105	132.768479, 46.215238	*F. sporotrichioides*	OM985219	OP436007	OP642186
SYS-51	132.768479, 46.215238	*F. sporotrichioides*	OM985220	OP436006	OP642187
HG-y102	130.440826, 47.312952	*F. sporotrichioides*	OM985121	OP436005	OP642188
HG-DBy101	130.440826, 47.312952	*F. sporotrichioides*	OM985122	OP436004	OP642189
SH-z61	127.270457, 46.64457	*F. acuminatum*	OM985115	OP435923	OP642072
SH-61	127.270457, 46.64457	*F. acuminatum*	OM985116	OP435922	OP642073
SH-41	127.270457, 46.64457	*F. acuminatum*	OM985117	OP435924	OP642074
HA-a72	126.868024, 45.850128	*F. acuminatum*	OM985221	OP435925	OP642075
HA-a161	126.868024, 45.850128	*F. acuminatum*	OM985222	OP435926	OP642076
HA-a162	126.868024, 45.850128	*F. acuminatum*	OM985223	OP435927	OP642077
HA-a163	126.868024, 45.850128	*F. acuminatum*	OM985224	OP435928	OP642078
HA-a164	126.868024, 45.850128	*F. acuminatum*	OM985225	OP435929	OP642079
HA-a1211	126.868024, 45.850128	*F. armeniacum*	OM985134	OP435979	OP642214
HA-13	126.868024, 45.850128	*F. armeniacum*	OM985135	OP435978	OP642215
HA-a121	126.868024, 45.850128	*F. armeniacum*	OM985136	OP435976	OP642216
HA-a122	126.868024, 45.850128	*F. armeniacum*	OM985137	OP435977	OP642217
HL-42	132.943466, 45.768947	*F. verticillioides*	OM985139	OP435994	OP642190
DQ-n32	125.835845, 46.329205	*F. verticillioides*	OM985141	OP435995	OP642191
SH-n12	127.270457, 46.64457	*F. verticillioides*	OM985142	OP435996	OP642192
JX-123	132.477436, 46.339951	*F. verticillioides*	OM985181	OP435997	OP642193
SH-n11	127.270457, 46.64457	*F. verticillioides*	OM985187	OP435998	OP642194
SH-n201	127.270457, 46.64457	*F. verticillioides*	OM985188	OP435999	OP642195
SH-n202	127.270457, 46.64457	*F. verticillioides*	OM985189	OP436000	OP642197
SH-n203	127.270457, 46.64457	*F. verticillioides*	OM985190	OP436001	OP642197
SH-n204	127.270457, 46.64457	*F. verticillioides*	OM985191	OP436002	OP642199
SH-n205	127.270457, 46.64457	*F. verticillioides*	OM985192	OP436003	OP642199
JX-3352	132.477436, 46.339951	*F. glycines*	OM985193	OP435937	OP642080
JX-335	132.477436, 46.339951	*F. glycines*	OM985194	OP435930	OP642081
HA-171	126.868024, 45.850128	*F. glycines*	OM985195	OP435936	OP642082
HA-172	126.868024, 45.850128	*F. glycines*	OM985196	OP435935	OP642083
HA-173	126.868024, 45.850128	*F. glycines*	OM985197	OP435934	OP642084
HA-174	126.868024, 45.850128	*F. glycines*	OM985198	OP435933	OP642085
WC-b53	127.22506, 44.93996	*F. glycines*	OM985208	OP435932	OP642086
HA-z1412	126.738196, 45.753014	*F. glycines*	OM985180	OP435931	OP642087
WC-22	127.22506, 44.93996	*F. annulatum*	OM985207	OP436048	OP642175
**NRRL 34034**	-	*F. ipomoeae*	**GQ505636**	**GQ505814**	-
**LC0455**	-	*F. ipomoeae*	**MK289580**	**MK289734**	-
**NRRL 45996**	-	*F. ipomoeae*	**GQ505671**	**GQ505849**	-
**CBS 140909**	-	*F. ipomoeae*	**MN170479**	**MN170412**	-
**NRRL 28029**	-	*F. compactum*	**GQ505602**	**GQ505780**	-
**NRRL 36318**	-	*F. compactum*	**GQ505646**	**GQ505824**	-
**NRRL 6548**	-	*F. flagelliforme*	**GQ505589**	**GQ505767**	-
**CBS 731.87**	-	*F. flagelliforme*	**GQ505600**	**GQ505778**	-
**LC12147**	-	*F. arcuatisporum*	**MK289584**	**MK289739**	-
**NRRL 32997**	-	*F. arcuatisporum*	**GQ505624**	**GQ505802**	-
**NRRL 45997**	-	*F. clavus*	**GQ505672**	**GQ505850**	-
**NRRL 34037**	-	*F. clavus*	**GQ505638**	**GQ505638**	-
**LC7937**	-	*F. citri*	**MK289640**	**GQ505816**	-
**LC7922**	-	*F. citri*	**MK289634**	**MK289788**	-
**NRRL 66939**	-	*Fusarium* sp.	**MW233217**	**MW233561**	-
**FRC R-9121**	-	*Fusarium* sp.	**MW233213**	**MW233557**	-
**CBS 462.94**	-	*F. sporotrichioides*	**MN120771**	**MN120750**	-
**NRRL 53430**	-	*F. sibiricum*	**HM744684**	**MW233474**	-
**NRRL 6227**	-	*F. armeniacum*	**HM744692**	**JX171560**	-
**FRC R-09335**	-	*F. armeniacum*	**GQ915501**	**GQ915485**	-
**NRRL 13818**	-	*F. asiaticum*	**AF212451**	**MW233412**	-
**NRRL 46738**	-	*F. asiaticum*	**FJ240299**	-	-
**NL19-100008**	-	*F. graminearum*	**MZ921906**	**MZ921775**	-
**CBS 136009**	-	*F. graminearum*	**MW928838**	**MW928826**	-
**NRRL 54216**	-	*F. acuminatum*	**HM068314**	**HM068334**	-
**JW 289003**	-	*F. acuminatum*	**MZ921908**	**MZ921777**	-
**CBS 130180**	-	*F. verticillioides*	**MW402024**	**MW402740**	-
**CBS 131389**	-	*F. verticillioides*	**MN534047**	**MN534288**	-
**CBS 135541**	-	*F. temperatum*	**MW402051**	**KU604284**	-
**CBS 130323**	-	*Fusarium* sp.	**MH485018**	**MH484927**	-
**CBS 214.49**	-	*F. glycines*	**MH484960**	**MH484869**	-
**CBS 127316**	-	*F. annulatum*	**MW402021**	**MW402738**	-
**CBS 100001**	-	*Macroconia leptosphaeriae*	**KM231959**	**HQ728164**	-

Bold accession numbers were generated from other studies.

**Table 2 jof-08-01170-t002:** Geographic origins and number of Fusarium isolates recovered from symptomatic maize leaves with macroscopic symptoms of leaf blight collected from 10 locations in Heilongjiang province, China.

Geographic Origins	Number of *Fusarium* Isolates										
	FIESC	*F. sporotrichioides*	*F. armeniacum*	*F. asiaticum*	*F. graminearum*	*Fusarium* sp.	*F. acuminatum*	*F. glycines*	*F. annulatum*	*F. temperatum*	*F. verticillioides*
Daqing city	2	0	0	0	0	0	0	0	0	0	1
Harbin city	56	0	4	1	2	3	5	5	0	1	0
Hegang city	0	3	0	1	1	0	0	0	0	0	0
Jixi city	5	0	0	0	0	0	0	2	0	0	1
Qiqihar city	1	0	0	0	0	0	0	0	0	0	0
Qitaihe city	1	0	0	0	1	0	0	0	0	3	0
Shuangyashan city	1	11	0	8	9	0	0	0	0	0	0
Suihua city	3	0	0	0	1	0	3	0	0	0	7
Hulin country	0	0	0	0	0	0	0	0	0	0	1
Wuchang city	1	0	0	0	0	0	0	1	1	0	0
Total	70	14	4	10	14	3	8	8	1	4	10
Percentage ^a^	47.95	9.59	2.74	6.85	9.59	2.05	5.48	5.48	0.68	2.74	6.85

^a^ Percentage = *n*/N × 100%, where *n* is the number of isolates for one species of Fusarium, and N is the total number of isolates for all Fusarium species.

## Data Availability

Sequences have been deposited in GenBank. The data presented in this study are openly available in NCBI. Publicly available datasets were analyzed in this study. These data can be found here: https://www.ncbi.nlm.nih.gov/, accessed on 3 September 2022.

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
