# Peer review of "Fusarium Species Associated with Maize Leaf Blight in Heilongjiang Province, China"

_jof, 2022, doi:10.3390/jof8111170_

Round 1

Reviewer 1 Report

The results can have potential, but some important aspects need to be deeply improved or explained:

-        The big problem is that the two genes tef1 and rpb2 are not enough to identify the fungi, especially fusarium, at the species level. Moreover, β-tubulin is more stable than rpb2 for fungal identification, so I suggest to the authors sequencing some representative fungal isolates with the β-tubulin gene to confirm fusarium identification.

Material and methods

-        2.1. Fusarium isolates collection: Preferred to mention the collecting location site (values of latitudes and longitudes)

-        2.3. DNA extraction and sequence analysis: what is the length of PCR amplification of tef1 and rpb2 genes

-        Table 1: what about the degree of sequencing similarity with other isolates or with the highest recorded isolate in the GenBank

Results

-        3.1. Fungal Isolation and Morphological Characterization: I think the morphological characteristics can’t groping the isolates 

-        Line 164-167: all fungal species must be italic. Please correct it throughout the manuscript

-        Line 103: “Gene sequences of 149 Fusarium isolates” where

Line 162: “In this study, 146 Fusarium isolates were obtained from symptomatic maize leaves”.

Table 1 contains 146 accession numbers of Fusarium isolates. Please correct.

-        Line 169: I think Figure 1 is not informative and didn't add value, you can delete it

-        3.2. Phylogenetic analysis: why authors concatenated sequences (tef1 and rpb2 genes) of the representative Fusarium isolates, not all isolates, as shown in figure 4. The authors must analyze the concatenated phylogenetic tree on all isolates (based on tef1 and rpb2 genes as well as a new gene) and to good comparing between isolates. Figures 4 and 5 showed differentiation between the grouping of fungal isolates, so the new concatenated phylogenetic tree of all isolates must be generated.

-        Line 252-254: The average disease incidence and average disease index caused by Fusarium species ranged from 23.03%-74.25% and 51.85-85.12, respectively (Table 3). The authors must present an individual disease index and disease incidence for each isolate. The average value is not acceptable; you are targeted to differentiate and study the relationships among Fusarium species and clarify the diversity of Fusarium species associated with maize leaf blight.

Discussion

- Must be improved and modified based on new results that will be added.

Author Response

Dear editor,

Thank you for the valuable suggestions. We have revised the manuscript according to your suggestions.

Thank you very much for your kindness and help.

Sincerely yours,

Dr. Xi Xu and Prof. Junwei Zhao

Answer to the comments of Reviewer #1:

  1. The results can have potential, but some important aspects need to be deeply improved or explained: The big problem is that the two genes tef1 and rpb2 are not enough to identify the fungi, especially fusarium, at the species level. Moreover, β-tubulin is more stable than rpb2 for fungal identification, so I suggest to the authors sequencing some representative fungal isolates with the β-tubulin gene to confirm fusarium identification.

Thank you for the valuable suggestions. We have sequenced the β-tubulin gene sequences of all Fusarium isolates, and we added this information into this manuscript, please see Table 1. Furthermore, the tef1 gene can be used to discriminate Fusarium species at the species or subspecies level, and the rpb2 gene was also more informative and frequently employed, so they were recommended to be sequenced for Fusarium species identification. However, although the partial beta-tubulin gene has been used for some Fusarium identification, it was not universal informative within Fusarium (O'Donnell et al, 2022). Certainly, we also have sequenced the partial beta-tubulin gene to confirm the status of our isolates but did not provide phylogenetic tree base on beta-tubulin gene sequences because this gene was unable to effectively distinguish these isolates, especially for species complex. In addition, the concatenated gene sequences of tef1 and rpb2 were enough and could well distinguish these isolates.

O'Donnell, K.; Whitaker, B. K.; Laraba, I.; Proctor, R. H.; Brown, D. W.; Broders, K.; Kim, H. S.; McCormick, S. P.; Busman, M.; Aoki, T.; Torres-Cruz, T. J.; Geiser, D. M. DNA Sequence-Based Identification of Fusarium: A Work in Progress. Plant Dis. 2022. 106, 1597–1609.

  1. Material and methods Fusarium isolates collection: Preferred to mention the collecting location site (values of latitudes and longitudes)

Thank you for the valuable suggestions. We have added the latitude and longitude of sampling location into this manuscript, please see Table 1.

  1. DNA extraction and sequence analysis: what is the length of PCR amplification of tef1 and rpb2 genes.

Thank you very much. The lengths of tef1 gene sequences were 546-686 bp, the lengths of rpb2 gene sequences were 794-896 bp and the lengths of β-tubulin gene sequences were 332-356 bp.

  1. Table 1: what about the degree of sequencing similarity with other isolates or with the highest recorded isolate in the GenBank

Thank you very much. We have added the similarity information into the Table S1.

  1. Fungal Isolation and Morphological Characterization: I think the morphological characteristics can’t groping the isolates.

Thank you for the valuable suggestions. Although it is not reliable to identify Fusarium only by morphology, it is reliable to roughly classify Fusarium according to its morphological characteristics. In this study, morphological characteristics are only for reference, and the final identification results are based on phylogenetic analysis.

  1. Line 164-167: all fungal species must be italic. Please correct it throughout the manuscript.

Thank you for the valuable suggestions. We have checked and revised.

  1. “Gene sequences of 149 Fusarium isolates” where.

Thank you very much! In this study, a total of 146 Fusarium isolates were obtained. I made a mistake here because of my carelessness. I am so sorry for my carelessness.

  1. “In this study, 146 Fusarium isolates were obtained from symptomatic maize leaves”.

Thank you for the valuable suggestions. In this study, a total of 146 Fusarium isolates were obtained. I made a mistake here because of my carelessness. I am so sorry for my carelessness.

  1. Table 1 contains 146 accession numbers of Fusarium Please correct.

Thank you for the valuable suggestions. In this study, a total of 146 Fusarium isolates were obtained. I made a mistake here because of my carelessness. I am so sorry for my carelessness

  1. Line 169: I think Figure 1 is not informative and didn't add value, you can delete it

Thank you for the valuable suggestions. We have removed this figure in our manuscript.

  1. 2. Phylogenetic analysis: why authors concatenated sequences (tef1 and rpb2 genes) of the representative Fusarium isolates, not all isolates, as shown in figure 4. The authors must analyze the concatenated phylogenetic tree on all isolates (based on tef1 and rpb2 genes as well as a new gene) and to good comparing between isolates. Figures 4 and 5 showed differentiation between the grouping of fungal isolates, so the new concatenated phylogenetic tree of all isolates must be generated.

Thank you for the valuable suggestions. We have revised and provide a new phylogenetic tree based on the concatenated gene sequences (tef1 and rpb2) of all isolates. Please see Fig. 3. Furthermore, the tef1 gene can be used to discriminate Fusarium species at the species or subspecies level, and the rpb2 gene was also more informative and frequently employed, so they were recommended to be sequenced for Fusarium species identification. However, although the partial beta-tubulin gene has been used for some Fusarium identification, it was not universal informative within Fusarium (O'Donnell et al, 2022). Therefore, we did not provide phylogenetic tree based on beta-tubulin gene sequences and did not use this gene to form the concatenated gene sequences because this gene was unable to effectively distinguish these isolates, especially for species complex. In addition, the concatenated gene sequences of tef1 and rpb2 were enough and could well distinguish these isolates.

O'Donnell, K.; Whitaker, B. K.; Laraba, I.; Proctor, R. H.; Brown, D. W.; Broders, K.; Kim, H. S.; McCormick, S. P.; Busman, M.; Aoki, T.; Torres-Cruz, T. J.; Geiser, D. M. DNA Sequence-Based Identification of Fusarium: A Work in Progress. Plant Dis. 2022. 106, 1597–1609.

  1. Line 252-254: The average disease incidence and average disease index caused by Fusarium species ranged from 23.03%-74.25% and 51.85-85.12, respectively (Table 3). The authors must present an individual disease index and disease incidence for each isolate. The average value is not acceptable; you are targeted to differentiate and study the relationships among Fusarium species and clarify the diversity of Fusarium species associated with maize leaf blight.

Thank you for the valuable suggestions. We have provided the individual disease index and disease incidence for each isolate, please see the Table S2.

  1. Discussion Must be improved and modified based on new results that will be added.

Thank you very much. We have improved and modified. Please see discussion section.

Reviewer 2 Report

The manuscript entitled “Fusarium species associated with maize leaf blight in Heilongjiang province, China” provides a study of the Fusarium species involved in maize leaf blight disease in the Heilongjiang province of China, morphologically and molecularly identified and tested for their pathogenicity on maize seedlings. Being species belonging to FIESC the most prevalent, a further genetic analysis on the different haplotypes in FIESC population, has been carried out. The results obtained in this study show the presence of 14 different species causing maize leaf disease, identified for the first time in China, among which 4 belonging to FIESC. Among FIESC strains, 7 different haplotype indicate evolving population.

 The manuscript provides useful information about Fusarium species in maize, and it is generally well written, except discussion paragraph, that needs a wider revision, by rewriting it.

I suggest below some general and specific revisions.

 Line 14. Replace “one of” with “among”

Line 15. Replace “of” with “on”

Lines 16-17. Change the sentence in “Based on morphological characteristics and phylogenetic analysis on translation elongation factor (tef1) and second-largest subunit of RNA polymerase II (rpb2) genes, 146 Fusarium isolates were obtained and grouped into 14 Fusarium species, including F. ipomoeae…”

Lines 21 and 22. Delete “Of which” and “”in the present study”

Line 23. Replace “that“ with “an evolving occurrence of the”

Line 24. Delete “became more diverse”

Lines 24-29. Replace the sentences from “The typical” to “expansion” with “The typical symptoms observed on maize leaves were oval to long strip lesions, gray to dark gray or brownish red in the center and chlorotic area at the edge. Based on tef1 gene 7 haplotypes of FIESC were identified in Heilongjiang province, suggesting a population expansion.”

Line 39. Replace “a series of maize diseases” with “several diseases on maize”

Line 40. Replace “Maize” with “maize”

Lines 40-42. Change the sentence in “Fusarium verticillioides was the first pathogen, reported in 1968 [6], causing maize leaf blight, and the only reported one up to now.”

Lines 43-44. Replace “caused by Fusarium species was” with “is”

Lines 46-48. Replace the sentences from “Only” to “symptoms” with “Thus, the identification of the pathogens based only on disease symptoms in the field is difficult”

Line 57. Add here further information on the effectiveness to use the gene rpb2 together with tef1, being this gene important for Fusarium species identification. In this way the authors can provide reasons for the choice of this gene in their study

Line 72. It is better to write “there are few researches on”

Line 75. Add “, based on tef1 and rpb2 genes,” after “relationships”

Line 82. “countries”

Lines 99-101. Since after these revisions the genes are named in lines 16-17, in this sentence write “The sequences of tef1 and rpb2 genes were amplified….” and replace comma with “and” between the names of the primers.

Lines 103-104. Replace the sentence with “Sequences of 149 Fusarium isolates were searched against GenBank and FUSARIOID-ID database (www.fusarium.org) [21] by Basic Local Alignment Search Tool (BLAST) analysis and then deposited into the NCBI GenBank (Table 1).”

Lines 106-107. Add “List of” at the beginning of the caption and “used in this study” at the end.

Line 110. Replace “subjected to” with “compared to the sequences available in”

Lines 131-135. Rewrite the sentences as follows “Disease severity (DS) and disease incidence (DI) were assessed 14 days post inoculation. DS was measured based on a 0-9 scale described by Rafael et al. [26] and Xu et al. [25]: 0 (no visible symptoms),…” and so on. Write only “DI” and delete“Disease index” in line 135.

Line 138 and 139. Delete “the” in both the lines

Line 141. Replace “in” with “with”

Line 144. Replace “The” with “Each”

Line 162. Add “in China” after “maize leaves”

From Line 162 until the end of the manuscript: Use italics for the name of the species or the genus. Check in all the text (i.e. lines 176, 191, 194, 198, 200, 203, 206, 209, 212, 215, 218, 221, 223, 225, 228, 232, 243, 250, and so on) and see also captions of Figure 1, Table 2, Figure 2.

Line 176. Correct “specie” with “species”

Lines 223-224. Also the name of the genes has to be written in italics. Check in the whole text of the manuscript (see lines 224, 227 and so on). Rewrite the sentence in “The sequences of tef1 and rpb2 genes of all Fusarium isolates obtained in this study were searched against the FUSARIOID-ID database (www.fusarium.org) by using BLAST analysis.

Figure 4. My suggestion is to use lighter colours and bigger fonts to make the figure more readable.

Line 243, Delete “results”

Line 248. Replace “with” with “to”

Line 250. Replace “these Fusarium isolates” with “all the Fusarium species”

Line 254. Use approximate numbers: “from 23 to 74% and from 52 to 85%, respectively.”

Lines 262-271. I suggest to change the caption in: “Fig. 6. a) Leaf blight symptoms on maize leaves caused by Fusarium species in the field; b)-o) Typical symptoms observed in greenhouse on maize leaves after inoculation with: b) F. ipomoeae; c) F. compactum; d) F. flagelliforme; e) F. asiaticum; f) F. armeniacum; g) F. citri; h) F. sporotrichioides; i) Fusarium sp.; j) F. glycines; k) F. graminearum; l) F. annulatum; m) F. temperatum; n) F. verticillioides; o) F. acuminatum.

Line 272. Replace “index” with “incidence”.

Table 3. Put the species in order of severity of disease.

Line 287. Replace comma with “;”

Discussion. I suggest to change the order of the topics rewriting the Discussion paragraph.

As suggestion, the authors should report at the beginning the aim of the study (lines 311-312), followed by general results (lines 305-306). Then report the species identified and discussion about them (part from line 322 to line 329). To follow, report results on pathogenicity, improving the discussion by comparing data of Fusarium species with other published papers. Then continue with the predominance of FIESC species (lines 309-311) and the analysis of haplotypes. At the end, finish with conclusions.

Author Response

Dear editor,

Thank you for the valuable suggestions. We have revised the manuscript according to your suggestions.

Thank you very much for your kindness and help.

Sincerely yours,

Dr. Xi Xu and Prof. Junwei Zhao

Answer to the comments of Reviewer #2:

Thank you very much for your valuable suggestion.

  1. The manuscript entitled “Fusarium species associated with maize leaf blight in Heilongjiang province, China” provides a study of the Fusarium species involved in maize leaf blight disease in the Heilongjiang province of China, morphologically and molecularly identified and tested for their pathogenicity on maize seedlings. Being species belonging to FIESC the most prevalent, a further genetic analysis on the different haplotypes in FIESC population, has been carried out. The results obtained in this study show the presence of 14 different species causing maize leaf disease, identified for the first time in China, among which 4 belonging to FIESC. Among FIESC strains, 7 different haplotype indicate evolving population. The manuscript provides useful information about Fusarium species in maize, and it is generally well written, except discussion paragraph, that needs a wider revision, by rewriting it.

Thank you very much!

  1. Line 14. Replace “one of” with “among”

Thank you very much for your suggestion. We have revised.

  1. Line 15. Replace “of” with “on”

Thank you very much for your suggestion. We have revised.

  1. Lines 16-17. Change the sentence in “Based on morphological characteristics and phylogenetic analysis on translation elongation factor (tef1) and second-largest subunit of RNA polymerase II (rpb2) genes, 146 Fusarium isolates were obtained and grouped into 14 Fusarium species, including ipomoeae…”

Thank you very much for your suggestion. We have revised.

  1. Lines 21 and 22. Delete “Of which” and “”in the present study”

Thank you very much for your suggestion. We have revised.

  1. Line 23. Replace “that“ with “an evolving occurrence of the”

Thank you very much for your suggestion. We have revised.

  1. Line 24. Delete “became more diverse”

Thank you very much for your suggestion. We have revised.

  1. Lines 24-29. Replace the sentences from “The typical” to “expansion” with “The typical symptoms observed on maize leaves were oval to long strip lesions, gray to dark gray or brownish red in the center and chlorotic area at the edge. Based on tef1 gene 7 haplotypes of FIESC were identified in Heilongjiang province, suggesting a population expansion.”

Thank you very much for your suggestion. We have revised.

  1. Line 39. Replace “a series of maize diseases” with “several diseases on maize”

Thank you very much for your suggestion. We have revised.

  1. Line 40. Replace “Maize” with “maize”

Thank you very much for your suggestion. We have revised.

  1. Lines 40-42. Change the sentence in “Fusarium verticillioides was the first pathogen, reported in 1968 [6], causing maize leaf blight, and the only reported one up to now.”

Thank you very much for your suggestion. We have revised.

  1. Lines 43-44. Replace “caused by Fusarium species was” with “is”

Thank you very much for your suggestion. We have revised.

  1. Lines 46-48. Replace the sentences from “Only” to “symptoms” with “Thus, the identification of the pathogens based only on disease symptoms in the field is difficult”

Thank you very much for your suggestion. We have revised.

  1. Line 57. Add here further information on the effectiveness to use the gene rpb2 together with tef1, being this gene important for Fusarium species identification. In this way the authors can provide reasons for the choice of this gene in their study

Thank you very much for your suggestion. The tef1 gene can be used to discriminate Fusarium species at the species or subspecies level, and the rpb2 gene was also more informative and frequently employed, so they were recommended to be sequenced for Fusarium species identification. However, although the partial beta-tubulin gene has been used for some Fusarium identification, it was not universal informative within Fusarium (O'Donnell et al, 2022). The concatenated gene sequences of tef1 and rpb2 were enough and could well distinguish these isolates. Therefore, we selected these two genes.

O'Donnell, K.; Whitaker, B. K.; Laraba, I.; Proctor, R. H.; Brown, D. W.; Broders, K.; Kim, H. S.; McCormick, S. P.; Busman, M.; Aoki, T.; Torres-Cruz, T. J.; Geiser, D. M. DNA Sequence-Based Identification of Fusarium: A Work in Progress. Plant Dis. 2022. 106, 1597–1609.

  1. Line 72. It is better to write “there are few researches on”

Thank you very much for your suggestion. We have revised.

  1. Line 75. Add “, based on tef1 and rpb2 genes,” after “relationships”

Thank you very much for your suggestion. We have revised.

  1. Line 82. “countries”

Thank you very much for your suggestion. They are counties, all these counties were in China.

  1. Lines 99-101. Since after these revisions the genes are named in lines 16-17, in this sentence write “The sequences of tef1 and rpb2 genes were amplified….” and replace comma with “and” between the names of the primers.

Thank you very much for your suggestion. We have revised.

  1. Lines 103-104. Replace the sentence with “Sequences of 149 Fusarium isolates were searched against GenBank and FUSARIOID-ID database (www.fusarium.org) [21] by Basic Local Alignment Search Tool (BLAST) analysis and then deposited into the NCBI GenBank (Table 1).”

Thank you very much for your suggestion. We have revised.

  1. Lines 106-107. Add “List of” at the beginning of the caption and “used in this study” at the end.

Thank you very much for your suggestion. We have revised.

  1. Line 110. Replace “subjected to” with “compared to the sequences available in”

Thank you very much for your suggestion. We have revised.

  1. Lines 131-135. Rewrite the sentences as follows “Disease severity (DS) and disease incidence (DI) were assessed 14 days post inoculation. DS was measured based on a 0-9 scale described by Rafael et al. [26] and Xu et al. [25]: 0 (no visible symptoms),…” and so on. Write only “DI” and delete“Disease index” in line 135.

Thank you very much for your suggestion. We have revised.

  1. Line 138 and 139. Delete “the” in both the lines

Thank you very much for your suggestion. We have revised.

  1. Line 141. Replace “in” with “with”
  2. Thank you very much for your suggestion. We have revised.
  3.  
  4. Line 144. Replace “The” with “Each”

Thank you very much for your suggestion. We have revised.

  1. Line 162. Add “in China” after “maize leaves”

Thank you very much for your suggestion. We have revised.

  1. From Line 162 until the end of the manuscript: Use italics for the name of the species or the genus. Check in all the text (i.e. lines 176, 191, 194, 198, 200, 203, 206, 209, 212, 215, 218, 221, 223, 225, 228, 232, 243, 250, and so on) and see also captions of Figure 1, Table 2, Figure 2.

Thank you very much for your suggestion. We have revised.

  1. Line 176. Correct “specie” with “species”

Thank you very much for your suggestion. We have revised.

  1. Lines 223-224. Also the name of the genes has to be written in italics. Check in the whole text of the manuscript (see lines 224, 227 and so on). Rewrite the sentence in “The sequences of tef1 and rpb2 genes of all Fusarium isolates obtained in this study were searched against the FUSARIOID-ID database (www.fusarium.org) by using BLAST analysis.

Thank you very much for your suggestion. We have revised.

  1. Figure 4. My suggestion is to use lighter colours and bigger fonts to make the figure more readable.

Thank you very much for your suggestion. We have revised.

  1. Line 243, Delete “results”

Thank you very much for your suggestion. We have revised.

  1. Line 248. Replace “with” with “to”

Thank you very much for your suggestion. We have revised.

  1. Line 250. Replace “these Fusarium isolates” with “all the Fusarium species”

Thank you very much for your suggestion. We have revised.

  1. Line 254. Use approximate numbers: “from 23 to 74% and from 52 to 85%, respectively.”

Thank you very much for your suggestion. We have revised.

  1. Lines 262-271. I suggest to change the caption in: “Fig. 6. a) Leaf blight symptoms on maize leaves caused by Fusarium species in the field; b)-o) Typical symptoms observed in greenhouse on maize leaves after inoculation with: b) F. ipomoeae; c) F. compactum; d) F. flagelliforme; e) F. asiaticum; f) F. armeniacum; g) F. citri; h) F. sporotrichioides; i) Fusarium sp.; j) F. glycines; k) F. graminearum; l) F. annulatum; m) F. temperatum; n) F. verticillioides; o) F. acuminatum.

Thank you very much for your suggestion. We have revised.

  1. Line 272. Replace “index” with “incidence”.

Thank you very much for your suggestion. We have revised.

  1. Table 3. Put the species in order of severity of disease.

Thank you very much for your suggestion. We have deleted Table 3 according to another reviewer.

  1. Line 287. Replace comma with “;”

Thank you very much for your suggestion. We have revised.

  1. I suggest to change the order of the topics rewriting the Discussion paragraph.

As suggestion, the authors should report at the beginning the aim of the study (lines 311-312), followed by general results (lines 305-306). Then report the species identified and discussion about them (part from line 322 to line 329). To follow, report results on pathogenicity, improving the discussion by comparing data of Fusarium species with other published papers. Then continue with the predominance of FIESC species (lines 309-311) and the analysis of haplotypes. At the end, finish with conclusions.

Thank you very much for your suggestion. We have changed the order of the topics in discussion section and rewritten.